# Effect of Graphene Oxide on the Mechanical Properties and Durability of High-Strength Lightweight Concrete Containing Shale Ceramsite

**DOI:** 10.3390/ma16072756

**Published:** 2023-03-30

**Authors:** Xiaojiang Hong, Jin Chai Lee, Jing Lin Ng, Zeety Md Yusof, Qian He, Qiansha Li

**Affiliations:** 1Department of Civil Engineering, Faculty of Engineering, Technology and Built Environment, UCSI University, Kuala Lumpur 56000, Malaysia; 2Department of Civil Engineering, Faculty of Civil and Hydraulic Engineering, Xichang University, Xichang 615013, China; 3Department of Civil Engineering, Faculty of Civil Engineering and Built Environment, Universiti Tun Hussein Onn Malaysia, Parit Raja 86400, Malaysia

**Keywords:** HSLWC, mechanical properties, durability, microstructure, GO

## Abstract

An effective pathway to achieve the sustainable development of resources and environmental protection is to utilize shale ceramsite (SC), which is processed from shale spoil to produce high-strength lightweight concrete (HSLWC). Furthermore, the urgent demand for better performance of HSLWC has stimulated active research on graphene oxide (GO) in strengthening mechanical properties and durability. This study was an effort to investigate the effect of different contents of GO on HSLWC manufactured from SC. For this purpose, six mixtures containing GO in the range of 0–0.08% (by weight of cement) were systematically designed to test the mechanical properties (compressive strength, flexural strength, and splitting tensile strength), durability (chloride penetration resistance, freezing–thawing resistance, and sulfate attack resistance), and microstructure. The experimental results showed that the optimum amount of 0.05% GO can maximize the compressive strength, flexural strength, and splitting tensile strength by 20.1%, 34.3%, and 24.2%, respectively, and exhibited excellent chloride penetration resistance, freezing–thawing resistance, and sulfate attack resistance. Note that when the addition of GO was relatively high, the performance improvement in HSLWC as attenuated instead. Therefore, based on the comprehensive analysis of microstructure, the optimal addition level of GO to achieve the best mechanical properties and durability of HSLWC is considered to be 0.05%. These findings can provide a new method for the use of SC in engineering.

## 1. Introduction

Lightweight concrete (LWC), compared with normal-weight concrete (NWC), has been widely and prosperously developed in the past decades due to its lower density and higher thermal insulation performance [1]. LWC can reduce the weight of cement by up to 20% without affecting the required strength, thus contributing to saving raw materials and transportation costs [2]. In addition, it was reported that LWC could reduce thermal energy consumption by approximately 15% in order to achieve thermal comfort in European buildings [3]. At present, many researchers have further explored the potential of mechanical properties and durability of LWC in order to develop high-strength lightweight concrete (HSLWC) with excellent performance [4,5]. Aslam et al. found that using oil-palm-boiler clinker (OPBC) to replace different proportions of coarse aggregate produced lightweight concrete with a 28-day compressive strength of 47 MPa [6]. Kılıç et al. showed that using basalt pumice produced structural HSLWC with a 90-day compressive strength of 43.8 MPa [7]. Rossignolo et al. reported that using local lightweight aggregates in Brazil as coarse aggregates could manufacture HSLWC with a 28-day compressive strength of 53.6 MPa [8]. As the scope of application expands, HSLWC has gradually evolved from nonstructural materials to structural materials. Typically, HSLWC can be used as an efficient structural material to increase the number of stories in high-rise buildings, extend the span of bridges, and strengthen the corrosion resistance of offshore platforms [9,10]. Furthermore, many countries are investigating the production of HSLWC from a variety of construction waste and recycled aggregate, such as ceramicite, fly ash, OPBC, pumice stone, and geopolymers, which has been facilitated in terms of environmental protection, the economy, and sustainable development [11,12,13].

Natural resources such as river sand and stone have been excessively consumed, while the accumulation of large quantities of shale spoil has also produced ecological pollution in China. Therefore, the government requires considerable human and material resources for the disposal and recycling of shale spoil every year. Under the incentive of the increasing demand of the light aggregate industry and the updating of ceramsite production technology, shale spoil can be mass produced into shale ceramsite (SC) and shale pottery sand (SPS) with different particle size distribution sunder high-temperature calcination processing [14]. Based on the characteristics of an increased number of pores, lighter weight, and higher strength, SC is considered as a good lightweight aggregate for producing HSLWC [15]. On the one hand, the bond strength of the interface between cement slurry and SC observed by SEM is an important contribution to strengthening the mechanical properties and durability of HSLWC. On the other hand, the porous structure of SC is not conducive to the compactness of concrete, thus limiting the physical reinforcement ability. With the same ratio of sand, the higher the content of SC, the lower the ultimate compressive strength of LWAC [16]. Although adversely affected by the high water absorption of SC, the slump of HSLWC is satisfactory under the condition of a low water–binder ratio [17].

Increasing the compressive strength to 55 MPa and improving durability are challenging problems for HSLWC, which is fundamentally limited by the lightweight and porous characteristics of aggregate [18]. The use of steel fiber, carbon fiber, and resin fiber as additives to strengthen and toughen the aggregate is a common way to overcome these problems [19,20]. Adding an appropriate amount of nanosilica and fly ash resulted in a positive synergistic effect on the mechanical properties and durability of SC concrete [21]. Nevertheless, nanoscale pores and microcracks still nucleate and grow in the hydration reaction of cement mortar during long-term loading [22]. Graphene oxide (GO) is a unique two-dimensional nanosheet structure that has attracted great attention in the research to improve the performance of cement mortar and concrete [23,24]. At present, the main industrial GO preparation process involves extraction from graphene with the modified Hummer’s method, and then cooling and drying in a vacuum [25]. With the advances in and development of preparation technology, GO is bound to be mass manufactured at a lower fabrication cost. Additionally, previous studies have demonstrated that the amount of GO added to cement-based materials has been optimized to a lower dosage [26,27]. Therefore, it is economically and technically feasible for the addition of GO to strengthen the performance of concrete. In general, the mechanism through which GO enhances concrete performance is currently in an intense exploration stage, and two strongly supportive theories are emphasized as follows: (1) the formation of flower-like crystals to regulate hydration reaction [28]; (2) the provision of nanoscale filling to enhance compactness [29].

The application of GO in cement-based materials mainly focuses on workability and mechanical properties. The better adhesion between GO and cement mortar leads to the lower slump and shorter setting time for fresh concrete [30]. GO mixed with in different proportions in cement mortar results in different degrees of enhancement in the mechanical properties. The addition of 0.03% GO as the optimal dose greatly reduced the total porosity of cement mortar and correspondingly increased the compressive strength by more than 40% [31]. Simultaneously, using the same proportion of GO was the most effective in promoting the growth and regulation of flower-like crystals [32]. When the addition amount exceeded 0.04%, the enhancement effect was inhibited due to the agglomeration of GO [33]. Similar conclusions on mechanical properties have been confirmed by other researchers for the application of pervious concrete and ultra-high-performance concrete (UHPC) [34,35]. However, limited information has been presented on the influence of GO on durability. Yu et al. reported that the incorporation of GO had a favorable impact on the durability of UHPC prepared from recycled sand [36]. Zeng et al. experimentally found that GO extended the life of cement mortars by more than two times when the specimens were exposed to sulfate attack [37]. In general, the many research achievements regarding GO have mainly concentrated on cement mortar and UHPC [38,39], but few studies have been conducted to investigate the influence of GO on the mechanical properties and durability of HSLWC. Hence, it is imperative to further analyze the potential of GO to stimulate HSLWC with SC as aggregate, so as to provide practical reference for construction.

The aims of this study were to use SC as a coarse aggregate to produce grade 60 HSLWC and to design six groups of GO mixtures with different contents to investigate the reinforcement effect and determine the optimal GO content. The following three indicators were used for comprehensive verification: (1) mechanical properties (compressive strength, flexural strength, and splitting tensile strength) tests in accordance with the GB/T 50081-2002 procedure; (2) durability (chloride penetration resistance, freezing–thawing resistance, and sulfate attack resistance) tests in accordance with the GB/T 50082-2009 procedure; (3) microstructure characterization.

## 2. Materials and Methods

### 2.1. Materials

Ordinary Portland cement (P.0 42.5 R), as the binder in all mixtures, was produced from Xichang Aerospace Co., Ltd. (Xichang, China). Its apparent density, specific surface area, and 28-day compressive strength were 3080 kg/m^3^, 320 m^2^/kg, and 47.4 MPa, respectively. All aggregates were manufactured by Hubei Huiteng Aggregate Co., Ltd. (Yichang, China). Their physical parameters are listed in Table 1, and their appearance is shown in Figure 1. In addition, the cylinder compression strength of SC was 5.2 MPa. Specifically, it can be seen from Figure 2a that SC presented a porous honeycomb structure on the surface, as measured by SEM. The fineness modulus of SPS was 2.96, which meets the requirements for medium sand in JGJ52-2016. GO, purchased from Suzhou Tanfeng Technology Graphene Technology Co., Ltd. (Suzhou, China), presented as a brownish powder, and its property parameters are illustrated in Table 2. Figure 2b shows that GO was a nanoscale sheet with a typical fine and dense wrinkle morphology, as measured by SEM. Fourier infrared spectroscopy (FTIR) of GO is shown in Figure 3. The four stretching vibration peaks of 3200, 1724, 1617, and 1064 cm^−1^ confirmed the activity of the typical oxidation functional groups -OH, C=O, C=C, and C-O, respectively. These functional groups play an important role in improving the hydrophilicity of GO, thereby contributing to better dispersion in mixtures [40]. Grade 1 of fly ash (FA) with a specific surface area of 420 m^2^/kg and loss on ignition of 3.48% was selected as the mineral admixture. The polycarboxylate superplasticizer (PS) not only had a water reduction rate of 12% but was also used as an active agent for dispersing GO.

### 2.2. Mix Proportions

The control mix proportion (GO-0) with a high-strength grade of 60 was calculated according to the absolute volume method according to GB/T 31387-2015. The advantage of this method is that the slurry plays the core role in improving strength, and the lightweight aggregate plays an auxiliary role of filling and reducing the apparent density. The other five trial mix proportions, numbered as GO-2, GO-4, GO-5, GO-6, and GO-8, were prepared, in which the added GO content was 0.02%, 0.04%, 0.05%, 0.06%, and 0.08% (by weight of cement), respectively. According to the prior studies on replacing cementitious materials with mineral admixtures to further obtain better fluidity and durability [41], the FA used in this study had a reasonable setting of 20%. The mix proportion of the six mixes is listed in Table 3.

### 2.3. Preparation and Curing

According to our previous research [42], PS needs to be used as an active agent with GO to achieve uniform dispersion in water under ultrasonic vibration for 30 min to finally prepare GO solution (GOS). GOS dispersion appeared uneven and particles were suspended when the GO content exceeded 0.05%. It was recommended that the prepared GO be used immediately. Furthermore, SC was presaturated for 24 h before casting, followed by naturally drying its surface in open air for reducing water absorption of SC. The mixing procedure was as follows: First, put the SC and SPS into a paddle mixer and mix for three minutes. Second, add cement and FA and continue mixing for three minutes. Third, mix 70% GOS solution and stir for another three minutes. Finally, add the remaining GOS into the mixer and mix for two minutes until the composite is well blended.

Each specimen casting was loaded into the plastic mold and vibrated at low speed on a vibration table. The specimen wrapped with plastic film was demolded after curing at ambient temperature for 24 h, and then placed in a standard curing box for curing until the test age. All performance tests were carried out according to the curing regime after reaching the specified age. Detailed mixing procedures and experimental items are shown in Figure 4. The results of all properties are reported as the average of the three specimens.

### 2.4. Experimental Methodology

#### 2.4.1. Mechanical Properties

In order to evaluate the effect of GO on the growth trend in compressive strength, the specimens with sizes of 100 × 100 × 100 mm were cast, at the ages of 1 d, 3 d, 7 d, 28 d, and 56 d for testing on a servo pressure testing machine (YAD-2015), as shown in Figure 5a. In addition, specimens with sizes of 100 × 100 × 400 mm were used to test the flexural strength at the age of 28d on a servo universal testing machine (RE-8030), as shown in Figure 5b. Specimens with sizes of 100 × 100 × 100 mm were used to test the splitting tensile strength at the age of 28 days on a computer control pressure testing machine (WAW-1000), as shown in Figure 5c.

#### 2.4.2. Durability

The rapid chloride ion migration coefficient (RCM) method was used to determine the chloride penetration resistance of HSLWC. The test pieces were cut equally from the standard cylinder specimen with a diameter of 100 mm and a height of 100 mm when the curing age reached 28 days, and then subjected to vacuum saturation treatment prior to testing. During the test, the test pieces were fixed on an RCM test device (RCM-NTB), as shown in Figure 6a. The voltage, average temperature, and duration were set to 30 V, 24 °C, and 24 h, respectively. Finally, the test piece, which was divided into two parts along the diameter direction, was used to measure the chloride intrusion depth on the cutting surface with a 0.1 mol/L AgNO_3_ solution. The chloride ion migration coefficient (DRCM) (Appendix A) is a quantitative index used to evaluate the permeability resistance.

In this study, the quick-freezing method with water freezing and water melting as the test environment was adopted, and specimens with sizes of 100 × 100 × 400 mm were cast for testing every 25 cycles. Before cycling, the specimen was first incubated for 24 days in standard curing and then immersed in water with the temperature of (20 ± 2) °C for 4 days in freeze–thaw testing machine. The test was designed for a total of 250 cycles; each cycle took 4 h; at the highest and lowest temperatures were −18 °C and 5 °C, respectively. The mass loss rate (Appendix A) and relative elastic modulus (Appendix A) were specified as quantitative indices to evaluate the frost resistance of HSLWC. The freeze–thaw testing machine (JCD-40S) and dynamic elastic modulus testing device (NELD-DTV) are shown in Figure 6b,c.

Specimens with sizes of 100 × 100 × 100 mm were prepared to measure the resistance to sulfate attack under wet and dry cycles. After 26 days of standard curing, all specimens were moved to an oven with a temperature of 80 ± 5 °C to dry for 2 days. In principle, the volume content of 5% Na_2_SO_4_ aqueous solution should reach at least 20 mm to the top surface of the topmost specimens, which can provide a good erosion environment. The stages and time of each cycle were as follows: we placed the specimens in solution (10 min), soaked (15 h), discharged the solution (15 min), air dried (1 h), oven dried (6 h), and then cooled the specimen (2 h). The test data were collected once every 30 cycles, for a total of 150 cycles. The mass loss rate (Appendix A) and corrosion resistance coefficient (Appendix A) were used as quantitative indicators of the sulfate resistance of HSLWC. The sulfate attack testing machine (NELD-VS830) is shown in Figure 6d.

#### 2.4.3. Microstructure

It was necessary to accurately analyze the mechanism of GO application in HSLWC. The microscopic characteristics of blocks with sizes of approximately 10 × 10 × 5 mm, which were taken from each group of mix proportion specimens at an age of 28 days, were observed from through SEM after gold-spray treatment.

## 3. Results and Discussion

### 3.1. Mechanical Properties

#### 3.1.1. Compressive Strength

Table 4 lists in detail the oven-dry density and compressive strength of specimens with different GO contents at 28 days under standard curing. All specimens had an oven-dry density in the range of 1696–1728 kg/m^3^ and a compressive strength in the range of 61.88–74.32 MPa, whereas the classic HSLWC has a density of no more than 1850 kg/m^3^ and a compressive strength of 35–79 MPa according to Monteiro et al. [43]. The specimens with GO (GO-5) obtained the highest compressive strength of 74.32 MPa, presenting a maximum increase of 20.1% compared with that of those without GO (GO-0). In addition, the specific strength (C/D in Table 4), defined as the ratio of compressive strength to density, is an important characteristic in assessing structural performance. The specimens without GO obtained a specific strength of 36.5 kN·m/kg, while the specimens with a GO content of 0.05% obtained a higher specific strength of 43.3 kN·m/kg. Shafigh et al. reported that the LWC produced with OPBC as coarse aggregate had a specific strength of 30.9 kN·m/kg [44]. Evangelista et al. found that HSLWC containing expanded shale had a specific strength of 36.3 kN·m/kg [45].

Figure 7 illustrates the compressive strength results of specimens with different GO incorporation contents at 1, 3, 7, 28, and 56 days under standard curing. Evidently, the specimens incorporating GO had a higher compressive strength than those without GO at each age, indicating that GO produced a positive and efficient contribution. However, the compressive strength showed a nonlinear trend of first increasing and then decreasing with the increase in GO content, suggesting that there is an optimal amount of GO to achieve maximum strength. It was noted that the optimum amount in this study was 0.05% of the mix (GO-5). Similar conclusions were drawn from the application of GO in UHPC according to Chu et al. and Yu. et al. [34,36]. The 3-day compressive strengths of GO-4, GO-5, GO-6, and GO-8 were significantly higher than the 7-day compressive strength of GO-0, and the 7-day compressive strength of GO-5 was very close to the 28-day compressive strength of GO-0, indicating that GO accelerated strength formation and thus shortened the curing time. This advantage provides economic convenience for the acceleration of template turnover and the reduction in construction costs [46].

#### 3.1.2. Flexural Strength

Figure 8a shows the flexural strength results of specimens with different GO incorporation contents at 28 days under standard curing. It was found that the flexural strength of the specimens incorporating GO was significantly higher than those without GO, which was attributed to the enhancement provided by GO. With the increase in GO content from 0 to 0.08%, the flexural strength presented a strong parabola trend. The flexural strengths of GO-0, GO-2, GO-4, GO-5, GO-6, and GO-8 were 6.47, 7.23, 8.11, 8.69, 8.34, and 8.25 MPa, respectively. Compared with GO-0, the flexural strengths of GO-2, GO-4, GO-5, GO-6, and GO-8 increased by 11.7%, 25.3%, 34.3%, 28.9%, and 27.4%, respectively. This result indicated that the optimal content of GO added to HSLWC was 0.05% in terms of improving the flexural strength.

Generally, the flexural strength/compressive strength ratio of HSLWC was around 9–10%, which is lower than that of NWC [47]. Meanwhile, the HSLWC with sintered fly ash obtained a flexural strength/compressive strength ratio of 20% due to modification with steel fiber [48]. The flexural strength/compressive strength ratio (F/C in Table 4) of all mixtures in this study was within the range of 5.9–6.7%. Relevant studies showed that GO had significant strengthening reinforcement effects on flexural strength compared with compressive strength in cement-based materials [28]. This indicated that the improvement in flexural strength was weaker than that in compressive strength, which might have been caused by the brittleness of SC. Figure 8b shows a necessary fitting relationship between compressive strength and flexural strength of HSLWC, and it is recommended that the equations from CEB-FIP for LWC be used for prediction with an estimated error of no more than 10%.

#### 3.1.3. Splitting Tensile Strength

Figure 9a presents the splitting tensile strength of HSLWC specimens with various contents of GO at 28 days under standard curing. Similar to flexural strength, the splitting tensile strength of the HSLWC with GO was evidently higher than that of those without GO. The variation trend in splitting tensile strength with the increase in GO content was similar to that of compressive strength and flexural strength. The splitting tensile strengths of GO-0, GO-2, GO-4, GO-5, GO-6, and GO-8 were 4.21, 4.65, 4.92, 5.23, 5.13, and 4.95 MPa, respectively. As expected, the specimens with GO achieved a maximum increase of 24% when the content of GO reached 0.05%.

The splitting tensile strength/compressive strength ratio of HSLWC was approximately 6.2–7.1%, while the range for NWC was 8~14% [49]. Lee et al. reported that the ratio of OPBC HSLWC after 28 days of full water curing was 6–7.7% [10]. The splitting tensile strength/compressive strength ratio (S/C in Table 4) of all mixtures in this study was within the acceptable range of 6.8–7.2%. Figure 9b shows a reasonable fitting relationship between compressive strength and splitting tensile strength of HSLWC. Various fitting equations for the prediction of splitting tensile strength are presented. Based on the test results of this study, the equation proposed in the literature obtained a minimum estimation error of less than 8% [50].

### 3.2. Durability

#### 3.2.1. Chloride Penetration Resistance

Figure 10 shows the chloride-ion migration coefficient results of specimens with different GO incorporation contents at 28 days. Similarly, the specimens incorporating GO obtained lower chloride-ion migration coefficients than those without GO, indicating that incorporating GO into HSLWC significantly promoted the resistance of chloride penetration. With the increase in GO content, the chloride-ion migration coefficients showed a nonlinear trend of first decreasing and then increasing, and the chloride-ion migration coefficient reached a minimum when the GO content was 0.05. This optimal GO content is consistent with the findings of a previous studies by Yu et al. [36]. The chloride-ion migration coefficient values of GO-0, GO-2, GO-4, GO-5, GO-6, and GO-8 were 7.18 × 10^−12^, 5.65 × 10^−12^, 4.44 × 10^−12^, 4.07 × 10^−12^, 4.25 × 10^−12^, and 4.91 × 10^−12^ m^2^/s, respectively, which suggested that GO can improve the chloride-ion migration coefficient of HSLWC to achieve a maximum reduction of 43%. The inherent key to the resistance of chloride-ion permeability is the pore size of concrete [51]. As mentioned earlier, owing to the filling of nanoscale particles, GO not only refines the structure to form harmless pores but also blocks or cuts off transportation to reduce porosity. On the contrary, GO gradually becomes agglomerated after the addition of more than the optimal content, which, in turn, lead to the deterioration of chloride-ion penetration resistance.

In addition, based on the resistance level divided by the 28-day chloride-ion migration coefficient, Luping et al. found that concrete with a migration coefficient greater than 18 × 10^−12^ had poor resistance to marine environments, and concrete with a migration coefficient less than 8 × 10^−12^ had good resistance to natural environments [52]. GO can provide better corrosion resistance for HSLWC to adapt to marine environments. Hence, it could be inferred that the migration coefficients of all mixtures in the study were within the satisfactory range.

#### 3.2.2. Freezing–Thawing Resistance

Figure 11 presents the freezing and thawing test results of specimens with different GO incorporation contents up to 250 days. As shown in Figure 11a, the mass loss rate of all mixtures gradually increased with the increase in cycles. Meanwhile, the mass losses of the specimens incorporating GO were lower than those of the specimens without GO, which indicated that GO could effectively slow down the mass losses of the during in the freezing and thawing cycles. After 250 cycles, the mass loss rates of GO-0, GO-2, GO-4, GO-5, GO-6, and GO-8 were 2.91%, 2.23%, 1.65%, 1.10%, 1.35%, and 1.92%, respectively. At this time, the surface of the specimens only grew some small holes but did not deteriorate to form cracks or peeling. The mass loss rate curve of GO-5 was higher than the mass loss rate curve of other mixes, indicating that 0.05% GO could reduce the mass loss rate to the greatest extent. As shown in Figure 11b, the relative dynamic elastic modulus had the same variation characteristics as the mass loss rate. After 250 cycles, the relative dynamic elastic moduli of GO-0, GO-2, GO-4, GO-5, GO-6, and GO-8 were 95.4%, 96.2%, 97.3%, 98.4%, 97.9%, and 96.7%, respectively. The dynamic elastic modulus curve of GO-5 was lower than the dynamic elastic modulus curve of other mixes, indicating that 0.05% GO could prevent the attenuation of the dynamic elastic modulus to the greatest extent.

The above results showed that 0.05% GO, as the optimal dosage, could help HSLWC obtain the best freezing and thawing resistance in this study. Freezing and thawing damage mainly depends on the swelling of capillary water in micro pores of concrete, whereas the addition of GO can refine pores and reduce porosity, thus impeding the free flow of capillary water. In addition, the mass loss rate and relative elastic modulus of all mixtures were in the range of 1.10–2.91% and 95.4–98.4%, respectively, which met the requirements of GB/T 50082-2009. With the contribution of GO, UHPC can obtain a relatively dynamic modulus in the range of 95.93–98.51% after 300 cycles [36]. It also should be highlighted that all the mixes in this study had excellent freezing and thawing resistance.

#### 3.2.3. Sulfate Attack Resistance

Figure 12 shows the sulfate attack resistance results of HSLWC with different GO incorporation contents up to 150 cycling times in a sulfate solution. As shown in Figure 12a, the mass of all mixtures slightly increased in the first 60 cycles and gradually decreased in the remaining 90 cycles, indicating that the mass loss ratio showed a trend of first negative growth and then positive growth. The specimens without GO had a more significant vibration amplitude in their mass change than those with GO, which was mainly due to the fact that GO prevented deterioration in the wet and dry cycles. After 150 cycles, the mass loss rates of GO-0, GO-2, GO-4, GO-5, GO-6, and GO-8 were 2.95%, 2.21%, 1.65%, 1.54%, 1.87%, and 1.99%, respectively. As shown in Figure 12b, the corrosion resistance coefficient had the same trend as the mass loss rate of first increasing and then decreasing. The corrosion resistance coefficient of the specimens without GO decreased after 60 cycles, while those of the specimens with GO decreased after 90 cycles. After 150 cycles, the corrosion resistance coefficients of GO-0, GO-2, GO-4, GO-5, GO-6, and GO-8 were 86.3%, 89.3%, 93.7%, 97.4%, 94.6%, and 91.3%, respectively. Accordingly, the nonlinear enhancement effect was probably attributable to the uneven dispersion or supersaturation of GO, suggesting that the optimal content for sulfate resistance was 0.05%.

Corrosion resistance was reported to be related to ion transport and pore structure [53]. At the early stage of corrosion, ettringite crystals were continuously formed and accumulated, filling capillary pores, thus temporarily increasing the compressive strength and weight. In the later stage of erosion, sulfate can gradually consume and destroy the skeleton of hydration products, thus reducing the compressive strength and weight. Considering the nanofold morphology, GO could block or cut off ion transport, thus mitigating corrosion damage [54]. Hence, all mixtures in this study had excellent corrosion resistance according to the requirements of GB/T 50082-2009.

### 3.3. Microstructure

Figure 13 shows the SEM images of samples randomly investigated from different mix proportions at 28 days. As shown in Figure 13a, some typical crystals were observed in the specimens without GO. These typical crystals were similar to those produced by hydration reaction in cement mortar, such as layered crystals, rod-like crystals, and sheet-like crystals, which were assembled from the composite formed by AFt, AFm, and CH. This process was also accompanied by the formation of nanoscale pores and microcracks.

The formation and growth of flower-like crystals could be observed from the samples containing GO, as shown in Figure 13b–f. In particular, in contrast with GO-0, many clusters of flower-like crystals formed and grew in an orderly manner in the interfacial transition zones and pores of the GO-2 mixture (Figure 13b). When the GO content increased from 0.04% to 0.06%, the petals of the flower-like crystals grew stronger and gradually matured (Figure 13c–e). When the content of GO reached 0.08%, the shape of the flower crystals were almost unchanged, and the number of flower crystals decreased (Figure 13f). Lv et al. confirmed that GO can participate in the hydration reaction to produce a unique and dense flower-like crystal [24]. Chuah et al. also reported that these flower-shaped crystals were beneficial to improving the mechanical properties and durability of concrete [55].

## 4. Conclusions

The main objective of this study was to design an initial mixture of HSLWC with SC as an aggregate, and we added six different low contents id GO to compare the enhancement effect of the mechanical properties and durability. In addition, the microstructure of HSLWC with different GO contents was also investigated. The main conclusions of this study are as follows:The specimens with different GO contents had an oven-dry density in the range of 1696–1728 kg/m^3^ and a compressive strength in the range of 61.88–74.32 MPa, which meet the classification requirements of HSLWC. GO not only adjusted the crystal morphology at an early stage but also maximized the 28-day compressive strength by 20.1%. The specimens with different GO contents had a flexural strength ranging from 6.47 to 8.69 MPa. The addition of GO could increase the flexural strength by 11.7–34.3%. The specimens with different GO contents had a splitting tensile strength ranging from 4.21 to 5.23 MPa. The addition of GO could increase the splitting tensile strength by 10.5–24.2%.The chloride-ion migration coefficient of HSLWC with different GO incorporation contents was within the range of 4.07 × 10^−12^–7.18 × 10^−12^ m^2^/s, suggesting that the HSLWC in this study could be well applied to marine environments. GO could help the chloride-ion migration coefficient of HSLWC to reach a maximum reduction of 43%. After 250 freezing and thawing cycles, the specimens with different GO contents had a mass loss rate in the range of 1.10–2.91% and a relative dynamic elastic modulus in the range of 95.4–98.4%. After 150 wet and dry cycles, the specimens with different GO contents had a mass loss rate in the range of 1.54–2.95% and a corrosion resistance coefficient in the range of 86.3–97.4%. These results indicated that GO can improve the freeze–thaw resistance and sulfate attack resistance of HSLWC.When the content of GO increased from 0 to 0.08%, all the performance indices of HSLWC showed a nonlinear trend. The peak in performance occurred when the GO content was 0.05%. It could be inferred that the optimal GO addition of HSLWC produced from SC was 0.05%. A low content of GO could adjust the crystal morphology to grow flower-like crystals. The number and size of flower-like crystals had a nonlinear relationship with the content of GO. This may be another important reason for the observed performance improvement.The results indicated that a low content GO could contribute better mechanical properties and durability to HSLWC, thereby extending the service life of buildings and reducing maintenance costs. The addition of different amounts of GO produces different reinforcement effects. GO can be used to achieve the application of SC in high-rise and large-span structures as well as in extreme cold or deep sea areas. Therefore, using GO to strengthen HSLWC made of SC has broad application prospects.Oxygen content is an important parameter for the affinity and mechanical properties of GO. Despite the significant mechanical and durability enhancements in this study, controlling the oxygen content of GO to accurately adjust the performance of HSLWC still requires further research to achieve wider practical applications.

## Figures and Tables

**Figure 1 materials-16-02756-f001:**
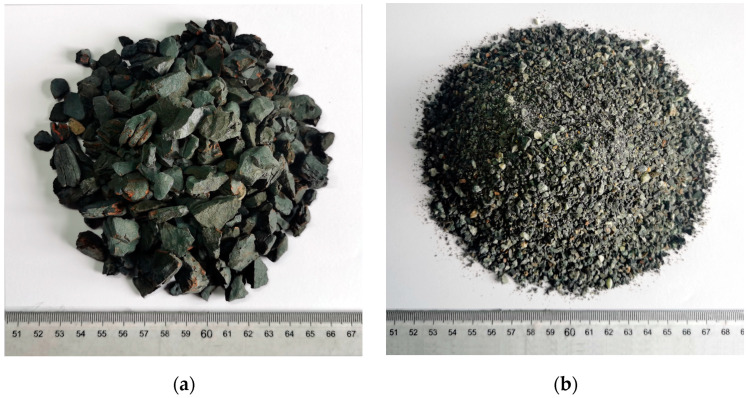
The images of aggregates: (**a**) SC; (**b**) SPS.

**Figure 2 materials-16-02756-f002:**
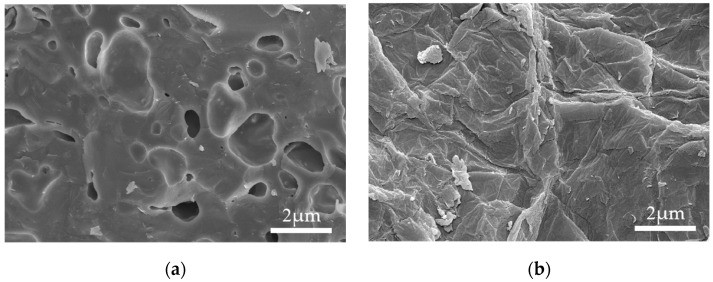
SEM images: (**a**) SC; (**b**) GO.

**Figure 3 materials-16-02756-f003:**
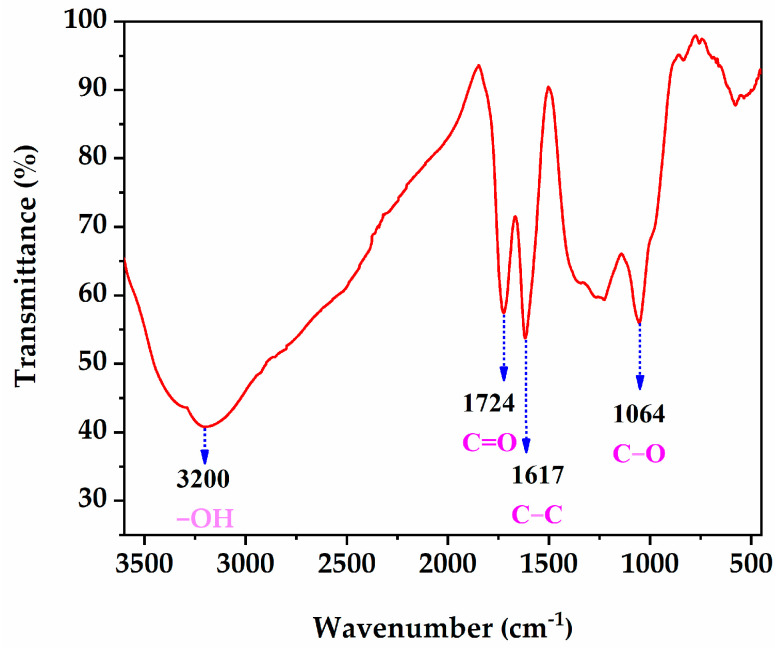
FTIR spectra of GO.

**Figure 4 materials-16-02756-f004:**
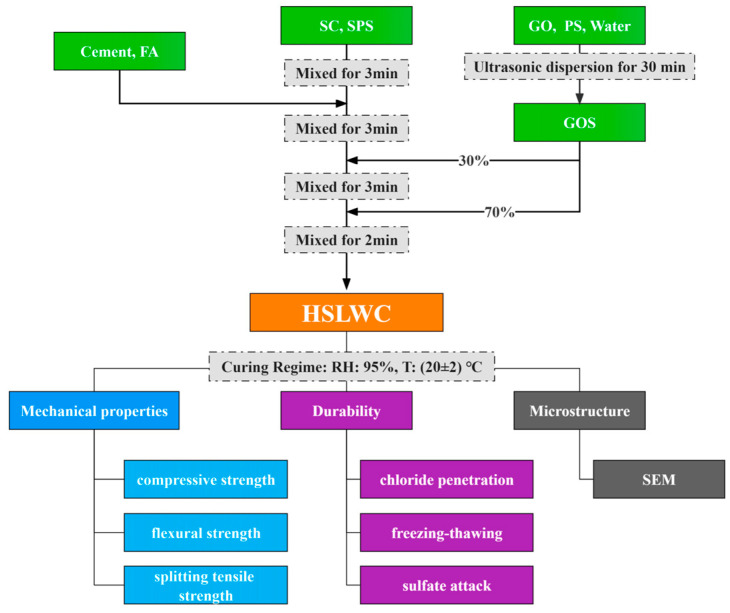
Mixing procedure and experimental items of HSLWC.

**Figure 5 materials-16-02756-f005:**
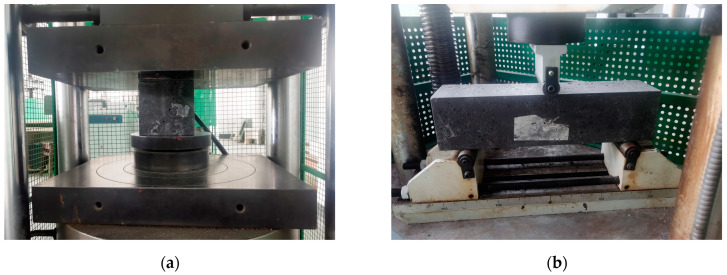
Test device for mechanical properties: (**a**) compressive strength test, (**b**) flexural strength test, and (**c**) splitting tensile strength test.

**Figure 6 materials-16-02756-f006:**
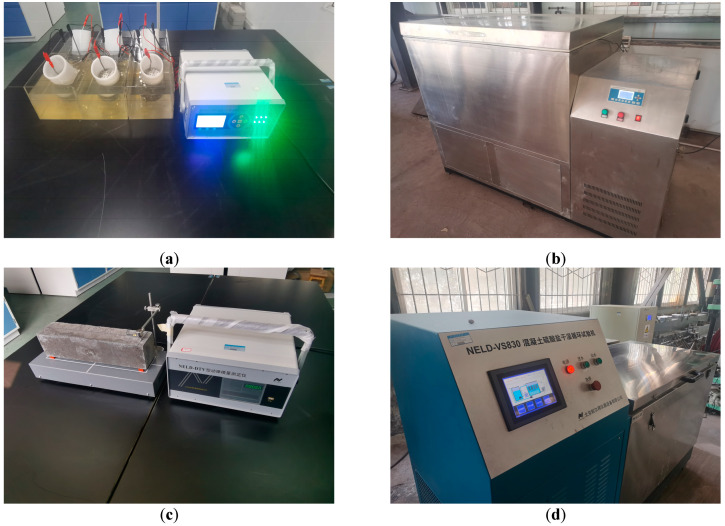
Test device for durability: (**a**) test device for rapid chloride ions migration (RCM) method, (**b**) freezing–thawing testing machine, (**c**) test device for dynamic modulus of elasticity, and (**d**) sulfate attack testing machine.

**Figure 7 materials-16-02756-f007:**
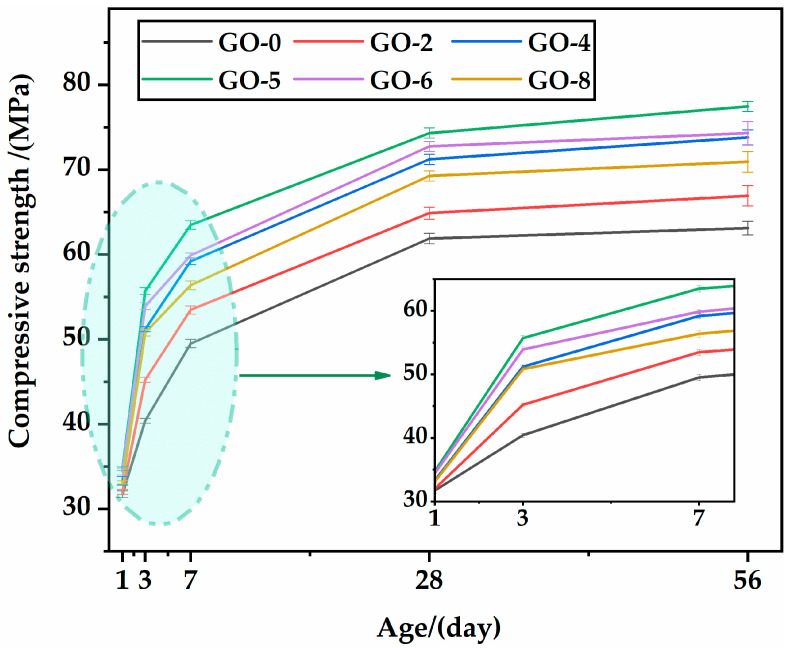
Compressive strength of HSLWC at different ages.

**Figure 8 materials-16-02756-f008:**
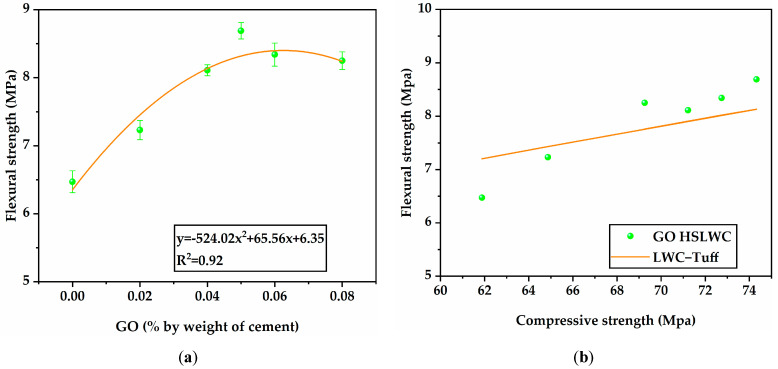
Flexural strength results of HSLWC: (**a**) flexural strength with different contents of GO; (**b**) prediction of flexural strength.

**Figure 9 materials-16-02756-f009:**
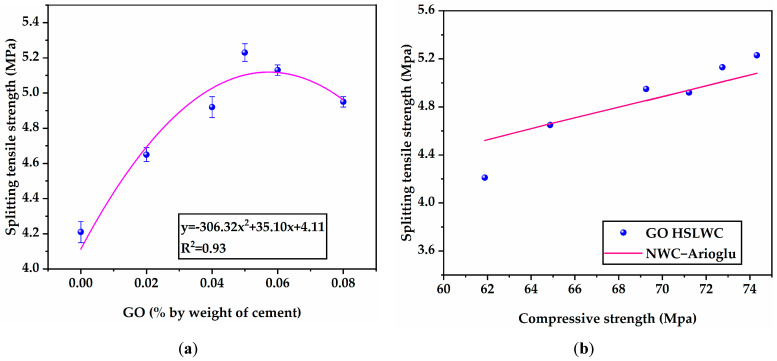
Splitting tensile strength results of HSLWC: (**a**) splitting tensile strength with different contents of GO; (**b**) prediction of splitting tensile strength.

**Figure 10 materials-16-02756-f010:**
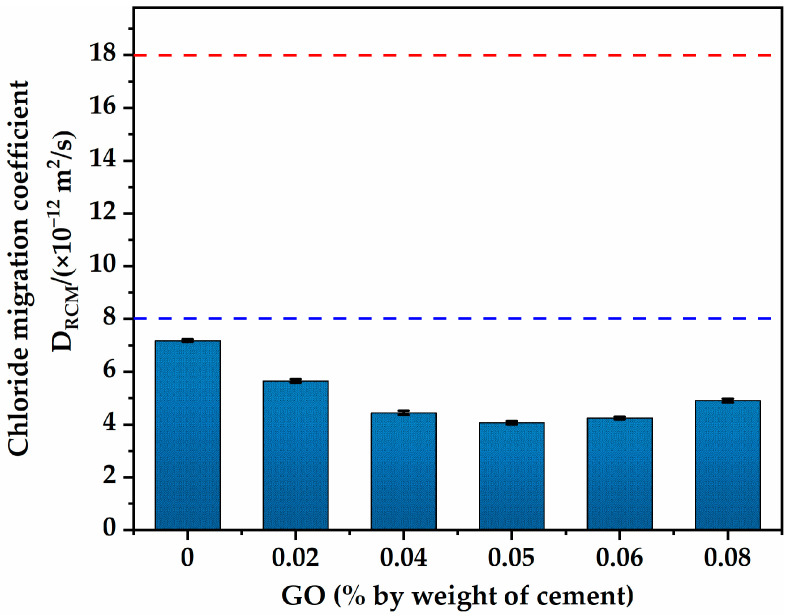
The chloride-ion migration coefficient of HSLWC with different contents of GO.

**Figure 11 materials-16-02756-f011:**
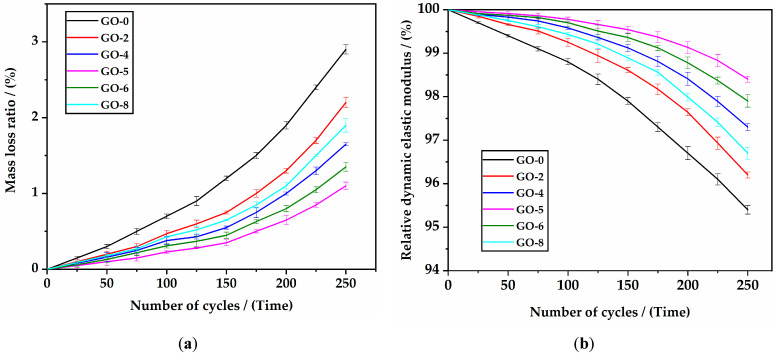
Freezing and thawing results of HSLWC with different GO incorporation contents: (**a**) the mass loss rate; (**b**) the relative dynamic elastic modulus.

**Figure 12 materials-16-02756-f012:**
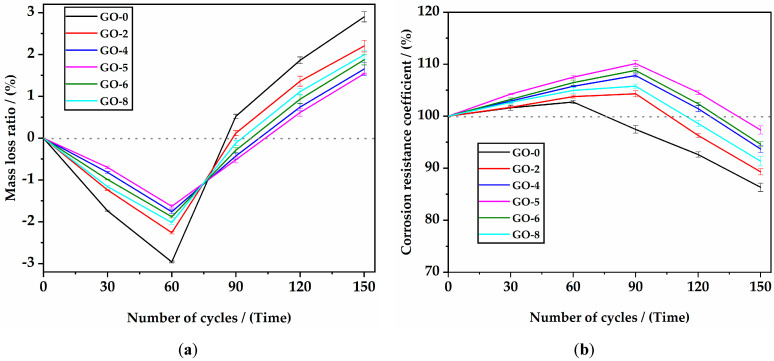
Sulfate attack resistance results of HSLWC with different GO incorporation contents: (**a**) mass loss ratio; (**b**) corrosion resistance coefficient.

**Figure 13 materials-16-02756-f013:**
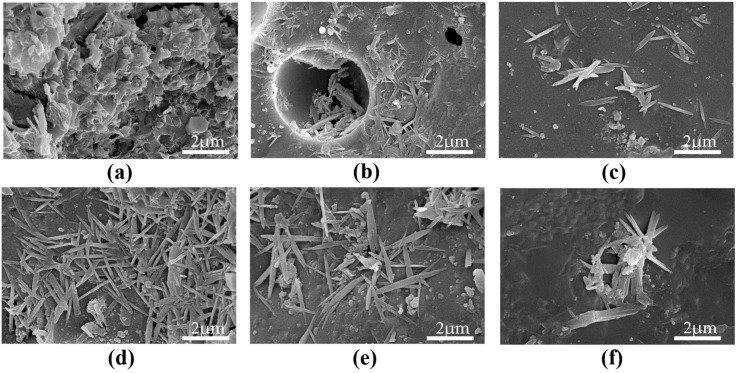
SEM images of different kinds of mix proportions at 28 days: (**a**) GO-0; (**b**) GO-2; (**c**) GO-4; (**d**) GO-5; (**e**) GO-6; (**f**) GO-8.

**Table 1 materials-16-02756-t001:** Physical properties of the aggregates.

Aggregate	Type	Density Rank (kg/m^3^)	Apparent Density (kg/m^3^)	Particle Size (mm)	Water Absorption (24 h) (%)
SC	coarse	800	1425	5–15	4.6
SPS	fine	700	1638	0–3	1.36

**Table 2 materials-16-02756-t002:** Property parameters of GO.

Specific Surface Area (m^2^/g)	Layers	Thickness (nm)	Diameter (µm)	Purity (%)	Oxygen Content (%)	Carbon Content (%)
100–300	1–2	~1	10–30	>95	>33	>66

**Table 3 materials-16-02756-t003:** Mix proportion (kg/m^3^).

No.	Cement	Water	SC	SPS	FA	PS	GO
GO-0	440	170	380	380	110	11	0
GO-2	440	170	380	380	110	11	0.088
GO-4	440	170	380	380	110	11	0.176
GO-5	440	170	380	380	110	11	0.220
GO-6	440	170	380	380	110	11	0.264
GO-8	440	170	380	380	110	11	0.352

**Table 4 materials-16-02756-t004:** The results and analysis of density and mechanical properties.

Mix No.	Density (kg/m^3^)	Compressive Strength (MPa)	C/D (kN·m/kg)	Ratio (%)	
				F/C	S/C
GO-0	1696	61.88	36.5	6.5	6.8
GO-2	1705	64.87	38.0	6.7	6.2
GO-4	1712	71.22	41.6	6.1	6.9
GO-5	1715	74.32	43.3	6.2	7.0
GO-6	1719	72.74	42.3	6.0	7.0
GO-8	1728	69.26	40.1	5.9	7.1

## Data Availability

Data are contained within the article.

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
