# Peer review of "Effect of Graphene Oxide on the Mechanical Properties and Durability of High-Strength Lightweight Concrete Containing Shale Ceramsite"

_materials, 2023, doi:10.3390/ma16072756_

Round 1

Reviewer 1 Report

GENERAL COMMENTS

1.    Standard deviation for all the test results must be given.

2.    Closing remarks can be added at the end of conclusion section based on the observations.

SPECIFIC COMMENTS

1.    Line 258: “This advantage provided economic convenience for the acceleration of template turnover and the reduction of construction costs”.

is use of graphene oxide comparatively reduces the construction cost?

2.    Line 279: “This indicates that the improvement of flexural strength was weaker than that of compressive strength, which might be caused by the brittleness of shale ceramsite.”

If the improvement of flexural strength of concrete depending on brittleness of shale ceramsite. What is the role of GO dosage?

3.    Line 323: “On the contrary, GO would gradually become agglomerated after the addition of more than the optimal content, which in turn leaded to the deterioration of chloride-ion penetration resistance.”

Is GO agglomeration identified?, what is the supporting observation for the statement

4.    Line 419: “GO can not only accelerate the hydration reaction at an early stage, but also maximize the 28 days compressive strength by 20.1%.”

Is hydration rate measured?, if yes include the results

Reviewer 2 Report

A primary objective of this study is to design the initial mixture of HSLWC and SC, and to add six different low content GO to compare the effect on mechanical properties and durability. While the topic of the paper is interesting, there are some suggestions to improve the quality of the paper:

1.  The introduction to a research paper is where you set up your topic and approach for the reader. It has several key goals: Present your topic and get the reader interested. Provide background or summarize existing research. Position your own approach. The authors should revise this section and also cite the following papers:

- https://doi.org/10.1016/j.jobe.2023.105905

- https://doi.org/10.3390/su142012990

- https://doi.org/10.1016/j.ceramint.2023.01.127

2. Clarification on the experimental procedure: The paper should provide a clear and detailed description of the experimental procedure used to prepare and test the high-strength lightweight concrete containing shale ceramsite. This should include information on the mixing process, curing conditions, and testing methods used to evaluate the mechanical properties and durability of the concrete.

3. More detailed analysis of the results: The paper should provide a more detailed analysis of the results obtained, including statistical analysis of the data. This would help to provide more insight into the effectiveness of graphene oxide in improving the mechanical properties and durability of the concrete.

4. Discussion of the limitations and future work: The paper should include a discussion of the limitations of the study and suggestions for future work. This could include recommendations for further research to better understand the mechanisms by which graphene oxide improves the properties of the concrete, as well as potential applications for the material in real-world construction projects.

5. Clarity on the significance of the results: The paper should provide a clear explanation of the significance of the results obtained. This should include an evaluation of how the findings contribute to the existing body of knowledge on high-strength lightweight concrete and how they could be applied in practice.

6. Suggestions for improving the presentation: Finally, the paper could benefit from some suggestions for improving the presentation, such as enhancing the quality of the figures and tables, or restructuring the text to improve clarity and flow.

Round 2

Reviewer 2 Report

I recommend that the editorial team consider publishing this manuscript.